# Prevalence and changing antimicrobial resistance profiles of *Shigella* spp. isolated from diarrheal patients in Kolkata during 2011–2019

Puja Bose[1], Goutam Chowdhury[1,2], Gourab Halder[1], Debjani Ghosh[1], Alok K. Deb[3], Kei Kitahara[2,4], Shin-ichi Miyoshi[2,4], Masatomo Morita[5], Thandavarayan Ramamurthy[1], Shanta Dutta[1], Asish Kumar Mukhopadhyay[1] *

1 Division of Bacteriology, ICMR-National Institute of Cholera and Enteric Diseases, Kolkata, India, 2 Collaborative Research Centre of Okayama University for Infectious Diseases at ICMR-NICED, Kolkata, India, 3 Division of Epidemiology, ICMR-National Institute of Cholera and Enteric Diseases, Kolkata, India, 4 Graduate School of Medicine, Dentistry and Pharmaceutical Sciences, Okayama University, Okayama, Japan, 5 Department of Bacteriology I, National Institute of Infectious Diseases, Tokyo, Japan

* asish_mukhopadhyay@yahoo.com

**Data Availability Statement:** All relevant data are within the manuscript and its Supporting information files. The sequences submitted in the

## Abstract

### Background

The primary aim of this study was to investigate the occurrence, characteristics, and antimicrobial resistance patterns of various *Shigella* serogroups isolated from patients with acute diarrhea of the Infectious Diseases Hospital in Kolkata from 2011–2019.

### Principal findings

During the study period, *Shigella* isolates were tested for their serogroups, antibiotic resistance pattern and virulence gene profiles. A total of 5.8% of *Shigella* spp. were isolated, among which *S. flexneri* (76.1%) was the highest, followed by *S. sonnei* (18.7%), *S. boydii* (3.4%), *and S. dysenteriae* (1.8%). Antimicrobial resistance against nalidixic acid was higher in almost all the *Shigella* isolates, while the resistance to *β-lactamases*, fluoroquinolones, tetracycline, and chloramphenicol diverged. The occurrence of multidrug resistance was found to be linked with various genes encoding drug-resistance, multiple mutations in the topoisomerase genes, and mobile genetic elements. All the isolates were positive for the invasion plasmid antigen H gene (*ipaH*). Dendrogram analysis of the plasmid and pulsed-field electrophoresis (PFGE) profiles revealed 70–80% clonal similarity among each *Shigella* serotype.

### Conclusion

This comprehensive long-term surveillance report highlights the clonal diversity of clinical *Shigella* strains circulating in Kolkata, India, and shows alarming resistance trends towards recommended antibiotics. The elucidation of this study's outcome is helpful not only in

GenBank are available with accession number PP196524 to PP196533.

**Funding:** This project was partially supported by the program of the Japan Initiative for Global Research Network on Infectious Diseases (J-GRID), JP23wm0125004 and JP23wm0225021 from Ministry of Education, Culture, Sports, Science and Technology in Japan (MEXT), and Japan Agency for Medical Research and Development (AMED). Grant recipient was SM. This research was also supported in part by Indian Council of Medical Research (ICMR), Govt. of India and the National Institute of Infectious Diseases (NIID), Japan (grant No.: JP23fk0108683 from AMED) to AKM. The funders had no role in study design, data collection and analysis, decision to publish, or preparation of the manuscript.

**Competing interests:** The authors have declared that no competing interests exist.

identifying emerging antimicrobial resistance patterns of *Shigella* spp. but also in developing treatment guidelines appropriate for this region.

## Author summary

*Shigella* spp. is one of the WHO-designated "antibiotic-resistant priority pathogens" that is the leading cause of bacterial diarrhea among children worldwide. The distribution of *Shigella* species varies depending on the geographic location, particularly in regions where environmental risk factors like substandard water quality, limited healthcare accessibility, and insufficient sanitation are widespread. The presence of serotype diversity and drug resistance combinations often leads to treatment failures. Therefore, it is crucial to comprehend the prevailing serogroups and antimicrobial resistance patterns of *Shigella* species in different geographical areas to implement effective treatment strategies. We have identified 535 *Shigella* positive cases from 9276 diarrheal cases. Multidrug-resistant (MDR) *Shigella* globally requires a profound understanding of its genetic diversity, as there is significant regional variation in MDR strains. In our study, we observed a broad spectrum of resistance among the *Shigella* strains, with over 53.6% of the isolates demonstrating multi-drug resistance. Representative isolates were evaluated for antimicrobial resistance encoding and virulence-associated genes as they enact significantly in disease development and interaction with the host immune system. This report marks the initial presentation of the outcomes from long-term surveillance and molecular characterization of antimicrobial resistances in clinical *Shigella* strains in Kolkata since 2011. The outcome of this investigations will guide in identifying emerging antimicrobial resistance trend, formulating suitable treatment guidelines tailored to the community's needs, and establishing baseline data for comparison with potential outbreak strains in the future.

## Introduction

*Shigella* spp. is one of the listed "antibiotic-resistant priority pathogens" that pose a significant risk to global health (WHO, 2017) and are responsible for acute invasive intestinal infection leading to dysentery characterised by watery or bloody diarrhea [1,2]. *Shigella* spp. (*S. dysenteriae*, *S. flexneri*, *S. boydii*, and *S. sonnei*) is the etiological agent of shigellosis, and its species distribution varies spatio-temporally. The frequency of isolation of *S. flexneri* is more in developing and less developed countries, whereas *S. sonnei* has been reported as a major threat in developed regions [3]. Both endemic and epidemic shigellosis are caused by many serotypes of *S. flexneri* and *S. dysenteriae* type 1 [4].

Globally, 165 million shigellosis cases have been reported every year in low- to middle-income countries, with more than 1 million deaths per year [5]. Access to clean water, good nutrition, sustained sanitation, and healthcare play a key role in reducing the disease burden [6,7]. *Shigella* has also been found to be the leading causative agent of childhood diarrhea, affecting 1.1 million children per year, predominantly infants to an age group below 5 years old [8].

In addition to the characteristic of a low infective dose, a major role in the pathophysiology of the disease and its interaction with the host immune system was played by the virulence characteristics of *Shigella* [9]. Amidst numerous well-known virulence factors associated with *Shigella* spp, invasion plasmid antigen (Ipa) and the invasion-associated locus (Ial), facilitate

intestinal cell penetration. The other factors, such as *Shigella* enterotoxin 1 (ShET-1) and *Shigella* enterotoxin 2 (ShET-2) are encoded by the chromosomally-located *set* gene, *set1*, and the plasmid-encoded gene *sen*, respectively [10]. Plasmid-derived VirF and VirB proteins control invasion-related gene transcription. *Shigella* spp. harbour toxic factors like serine protease autotransporters of Enterobacteriaceae (SPATE) that include class 1 toxins, *Shigella* IgA-like protease homologue (SigA), and secreted autotransporter toxin (Sat). The other is non-toxic SPATE class 2 toxins, which include the extracellular serine protease (*sepA*), whose association is mainly with intestinal inflammation, and colonisation [11]. Pulsed-field electrophoresis (PFGE) is the most popular technique used to distinguish clonal isolates among various molecular methods employed for strain typing [12].

Although *Shigella* seems to be a self-limiting disease, the WHO guidelines recommend antimicrobials in the treatment of the disease to control epithelial infiltration, duration of diarrhea, severity of symptoms, disease carrier stage, and death incidence [13]. The treatment of shigellosis has become more challenging due to the emergence of multidrug-resistant (MDR) strains. Consequently, MDR strains of *Shigella* are increasingly reported to be resistant to most of the recommended antimicrobials, like ciprofloxacin, third-generation cephalosporin, ceftriaxone, and macrolide azithromycin [8]. Current data suggests an annual global fatality rate of approximately 700,000 due to antimicrobial resistance (AMR) [5].

AMR is also reported to be associated with a variety of biological, pharmacological, and social variables, like this organism's ability to acquire extra genetic material in the form of plasmids, pathogenicity islands, transposons, etc., as well as indiscriminate antibiotic use [14–18]. Significant regional variation in MDR *Shigella* across the world makes it crucial to depend on information about its genetic diversity in determining epidemiological shifts as well as the development of appropriate treatment methods.

Different studies have been made across the world to track the epidemiology and antimicrobial susceptibility patterns of *Shigella* spp. [5,19–22]. MDR *Shigella* spp. has shown a rising trend in Kolkata, particularly in relation to the fluoroquinolone class of antimicrobials [23]. However, there has been a dearth of information related to prevalence and the AMR pattern in *Shigella* strains isolated from Kolkata and its surrounding areas since 2010. Hence, the present study aimed to determine the epidemiological pattern of shigellosis and its antimicrobial resistance (AMR) patterns in this particular region.

## Materials and methods

### 1. Ethics statement

Informed written consent was obtained from the eligible cases or their legal guardian after explaining to them the purpose of this study. The work carried out in this study was approved by the Institutional Ethical Committee (IEC) of ICMR-National Institute of Cholera and Enteric Diseases (NICED), Kolkata, India (approved no. A-1/2015-IEC).

### 2. Study design

From January 2011 through December 2019, 9276 enrolled patients with diarrhoea at Infectious Disease Hospital, Kolkata, were tested, and 535 (5.8%) were found to have *Shigella* spp. Stool samples were tested for common enteric pathogens using standard microbiological methods within 2 hours of collection [24]. Antisera from a commercial supplier (Denka Seiken Co. Ltd., Tokyo, Japan) were used in slide agglutination tests to verify the identification of *Shigella* spp. isolates based on biochemical methods. The isolates were further checked for their multidrug resistance pattern and the prevalence of the various antimicrobial resistance genes along with the virulence associated genes which would help in formulating better treatment

strategies for the community. Genetic relatedness of these isolates was also determined by analysing their PFGE patterns and plasmid profiling. This research had previously been approved by the institution's ethics review board. Each isolate was preserved in a 15% glycerol stock that was frozen at -80˚C.

## 3. Antimicrobial susceptibility testing and determination of MIC

*Shigella* isolates were tested for their susceptibility to various antibiotics using the Kirby-Bauer disc-diffusion method on Mueller-Hinton agar, as recommended by the Clinical and Laboratory Standards Institute [25]. Commercially available discs (Becton Dickinson, BBL): nalidixic acid (NA; 30μg), ciprofloxacin (CIP; 5μg), norfloxacin (NOR; 10μg), ofloxacin (OFX; 5μg), tetracycline (TET; 30μg), ampicillin (AM; 10μg), streptomycin (S; 10μg), trimethoprim-sulfamethoxazole (SXT; 1.25/23.75μg), ceftriaxone (CRO; 30μg) and chloramphenicol (CHL; 30μg) were used in this assay. Following the CLSI standards, we interpreted the data. For a subset of resistant isolates, we used the agar dilution method to determine the CLSI-recommended minimum inhibitory concentrations (MICs) of nalidixic acid, ciprofloxacin, ampicillin, tetracycline, streptomycin, and chloramphenicol. In order to ensure consistent results, we used *Escherichia coli* ATCC 25922 as a reference strain [25].

## 4. Preparation of template DNA

Bacterial cells that had been cultured for 24 hours in a volume of 200μl of TE buffer were heated to 100˚C for 10 minutes and then stored on ice. After the thermal shock, a centrifuge was used to spin the resulting suspension at 10,000 rpm for 5 minutes. The template DNA for the PCR assay came from about 4 μl of the supernatant.

## 5. PCR amplification of virulence genes

The genes for virulence (*ipaH*, *ial*, *set* (sHET-1), *sen* (sHET-2), *virF*, *virB*, *sat*, *sigA*, *sep*, and *pic*) were identified by polymerase chain reaction (PCR) with the primers provided in S1 Table.

## 6. Determination of integron and AMR gene markers by PCR

The three integrons (*intI*, *intII*, and *intIII*) were identified using PCR techniques. To further categorise the isolates into those with typical or atypical class 1 gene cassettes, we looked for the presence of the qacEΔ1 gene at the 3' end of the class 1 conserved segment and at the gene cassette within the variable region of class 1 and class 2 integrons. Direct sequencing of the PCR product was used to identify class 1 and class 2 resistance gene cassettes. The BLAST programme was used to compare the obtained sequences to those in the NCBI database, and the results were analysed accordingly.

In order to screen for the presence of specific resistance genes, PCR was performed on MDR *Shigella* isolates using previously published primers. (S2 Table). Several resistance genes like, β-lactamase genes (*bla*$_{OXA}$, *bla*$_{TEM}$, *bla*$_{CTX-M}$); sulphonamide (*sul2*) and trimethoprim (*dfr1*) resistance genes; chloramphenicol (*catA*) resistance gene; streptomycin (*aadA*, *strA*); tetracycline (*tetA*, *tetB*) were mainly screened. In addition, the presence of plasmid-mediated quinolone resistance (PMQR) genes (*qnrA*, *qnrB*, *qnrD*, *qnrS*, and *aac-(6')-1b*) was investigated in some quinolone-resistant isolates. The role of topoisomerase mutations in quinolone resistance was investigated by using two or more representative isolates with differing profiles of quinolone resistance. The mutations were found by amplification and sequencing of the

quinolone resistance determining regions (QRDRs) of topoisomerase (*gyrA*, *gyrB*, *parC*, and *parE*) and comparison to *Shigella* control strain DNA (Accession: NC_004741.1) from the database.

### 7. Plasmid profiling

Plasmids were extracted from representative isolates using a modified version of the Kado and Liu method [26]. Plasmid profiles were characterised by electrophoresis on an agarose gel with a dilution of 0.8%. After electrophoresis and ethidium bromide staining, the gels were photographed using a Bio-Rad gel documentation system. Using a DNA ladder as a standard, the migration patterns of the unknown plasmids were compared with those of the known plasmids, allowing their molecular weights to be calculated.

### 8. Analysis of similarity among isolates with the help of plasmid profiling

Dendrograms were constructed using the FPQuest software to show the degree to which the isolates were similar based on their plasmid profiles. The unweighted pair group method (UPGMA analysis) was applied to the matrix similarity coefficients (1.5% position tolerance) to produce a dendrogram with the average linkage method.

### 9. Molecular typing of isolates by PFGE

Following the PulseNet standard protocol (https://www.cdc.gov/pulsenet/pdf/ecoli-shigella-salmonella-pfge-protocol-508c.pdf), DNA fingerprinting was carried out by PFGE of *Xba*I digested genomic DNA of the study strains using CHEF DRIII (Bio-Rad). The molecular DNA ladder consisted of *Xba*I-digested *Salmonella enterica* serovar Braenderup H9812. Gel images were recorded using a Gel-Documentation system (Bio-Rad), and PFGE profiles were analysed with BioNumerics software (Applied Maths, version 5.0). Similarity between bands was measured using the Dice coefficient, and correlation coefficients between clusters were calculated using the UPGMA method.

### 10. Statistical analysis

Statistical analysis was performed using Chi-square test for infection rate, age distribution part and Chi-square test for trend was used for calculating trend analysis of seasonality and antimicrobial resistance trend by using STATA v.18.0 (StataCorp LLC., College Station, Texas). A *p* value of < 0.05 was considered statistically significant.

## Results

### 1. Prevalence of *Shigella* spp

During the study period, 535 (5.8%) *Shigella* spp. were isolated from 9,276 enrolled diarrheal cases. The macroscopic characteristics of stools related to Shigellosis have different characteristics like, majority of the infected stools had muciod consistency (48%), 31% was bloody diarrheal stool, 12% was watery diarrhoeal stool, whereas 4% and 5% were semisolid stool and swab respectively. The numbers and percentage of different *Shigella* serogroups isolated were 407 (76.1%) *S. flexneri*, 100 (18.7%) *S. sonnei*, 18 (3.4%) *S. boydii, and* 10 (1.8%) *S. dysenteriae*. Table 1 shows the yearly distribution of *Shigella*. The number of stool specimens collected each year started increasing in 2013 and remained same throughout the study period. The isolation rate of *Shigella* spp. remained relatively stable during the study period, but in 2013, the rate of isolation decreased drastically. In 2019, the *Shigella* infection rate reached its peak, while its lowest occurrence was observed in 2013. In fact, approximately 4-fold increase in

**Table 1. Yearly distribution and infection of *Shigella* serogroups from stool sample of acute diarrheal patients at ID hospital, 2011–2019.**

| Year | 2011 | 2012 | 2013 | 2014 | 2015 | 2016 | 2017 | 2018 | 2019 | TOTAL |
|---|---|---|---|---|---|---|---|---|---|---|
| *Total no. of stool specimens* | 643 | 967 | 1184 | 1135 | 1195 | 1266 | 1254 | 940 | 692 | 9276 |
| *S.dysenteriae* | 1(0.16) | 2(0.2) | 0(0) | 1(0.09) | 0(0) | 4(0.32) | 0(0) | 1(0.11) | 1(0.14) | 10(0.11) |
| *S.flexneri* | 31(4.82) | 58(5.99) | 21(1.77) | 35(3.08) | 46(3.84) | 47(3.71) | 71(5.66) | 48(5.11) | 50(7.23) | 407(4.39) |
| *S.boydii* | 0(0) | 5(0.52) | 2(0.17) | 5(0.44) | 0(0) | 1(0.08) | 1(0.08) | 1(0.11) | 3(0.43) | 18(0.19) |
| *S.sonnei* | 4(0.62) | 11(1.14) | 3(0.25) | 11(0.97) | 19(1.59) | 12(0.95) | 19(1.52) | 13(1.38) | 8(1.16) | 100(1.08) |
| *Total no. isolated* | 36(5.59) | 76(7.86) | 26(2.2) | 52(4.58) | 65(5.44) | 64(5.06) | 91(7.26) | 63(6.70) | 62(8.96) | 535(5.77) |

annual infection rate from 2.2% in 2013 to 9% in 2019 (Fig 1) was noted. There is a significant difference in the isolation rate (p < 0.00001). Fig 2 illustrates the overall monthly isolation of all *Shigella* spp. during 9 years, the trends in the *S. sonnei* isolation rate with relation to seasonality show a decreasing trend from 2012 onwards (S1 Fig), *i.e.*, the isolation rates were high during the pre-monsoon season (March–May) and low during the winter season (December–

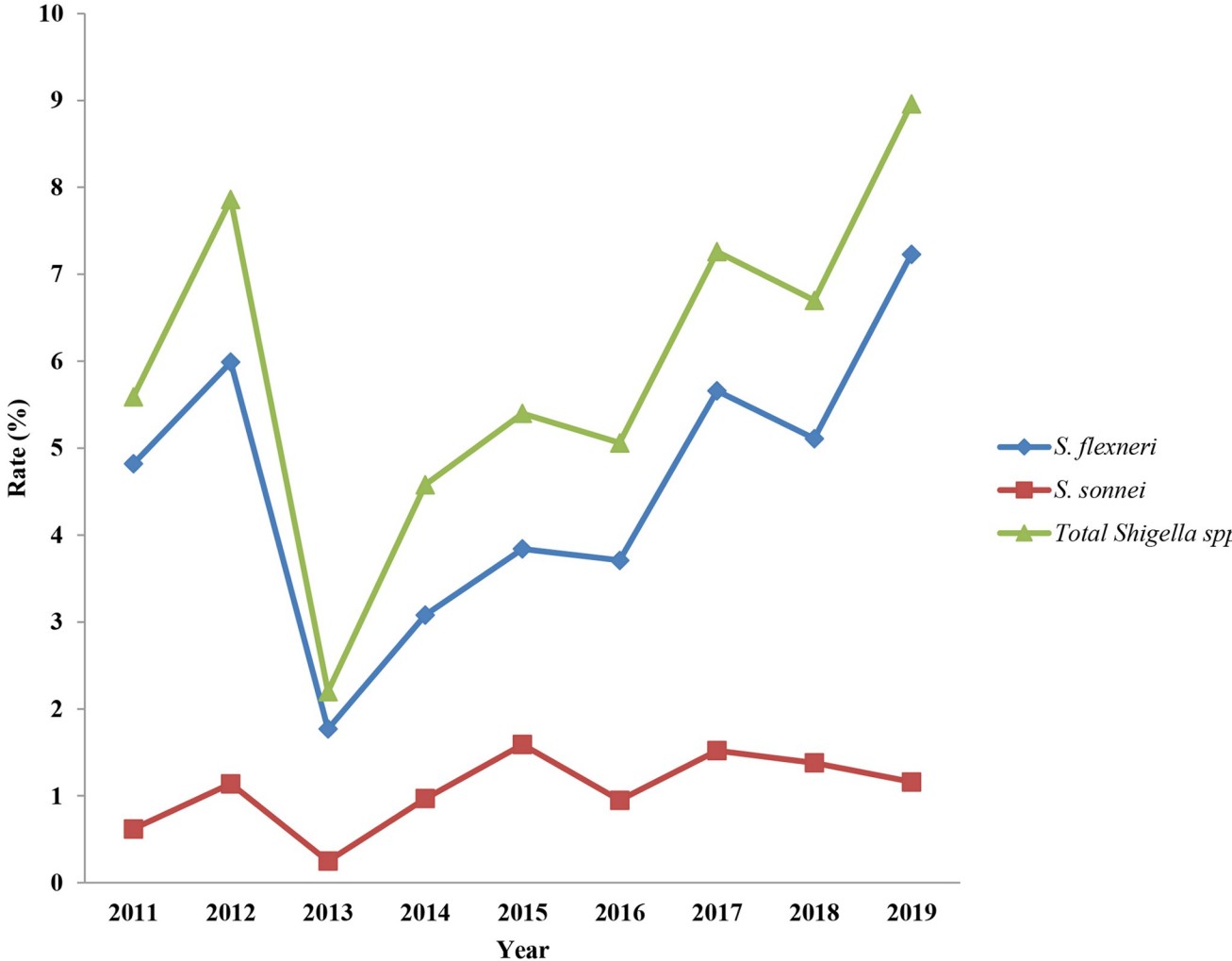

**Fig 1. Distribution of *Shigella* isolates in patients during 2011 and 2019 in Kolkata, India.** *Shigella* infection rate of (%): the proportion of shigellosis cases in diarrhea patients.

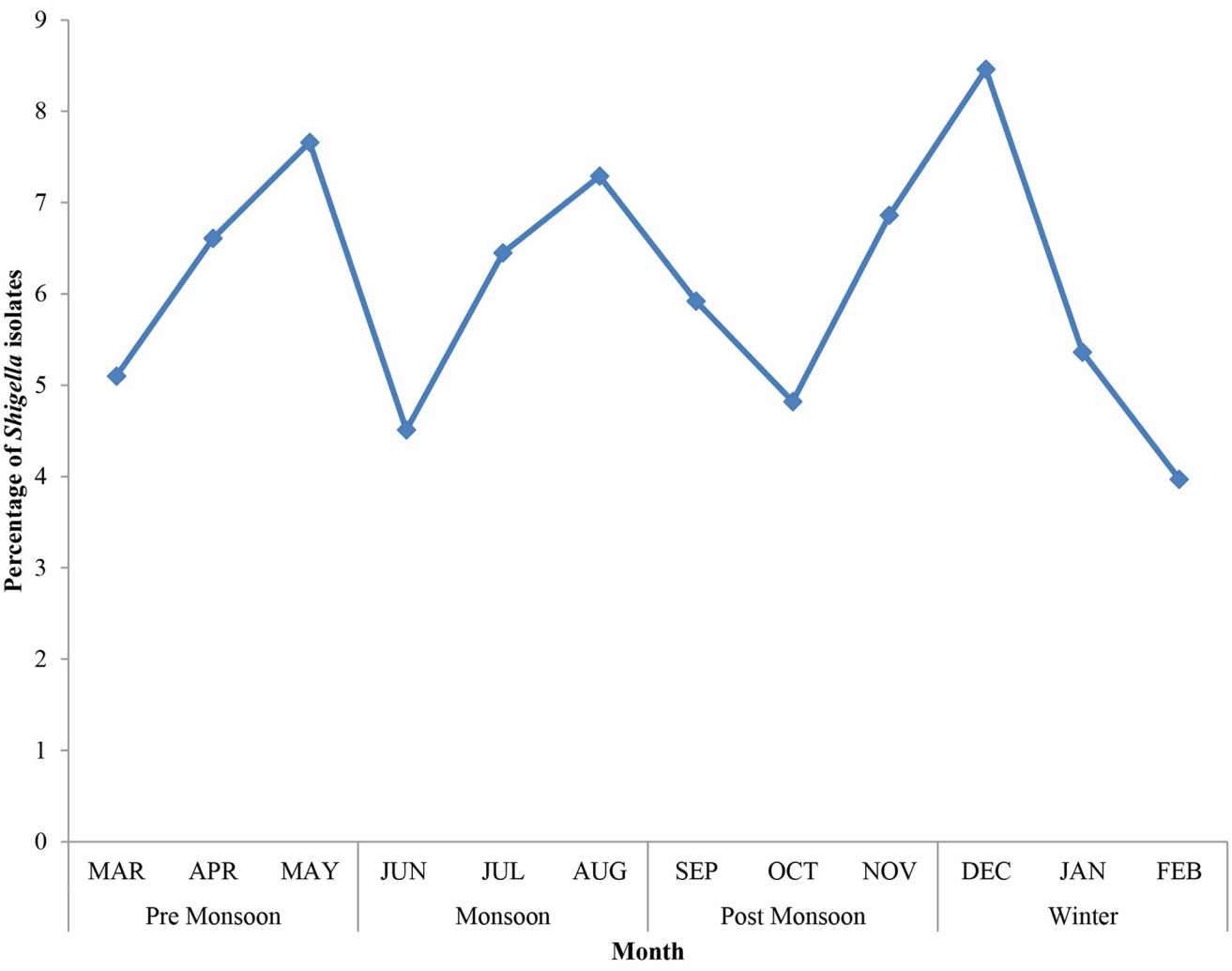

**Fig 2. Seasonal distribution of total shigellosis cases.**

February). *S. sonnei* showed a slightly reverse trend in the year 2011. No such trend was observed for *S. flexneri* (S2 Fig).

The proportion of *Shigella* cases varied majorly by age groups, with the highest rate being 9% in 6–14 years of age group, followed by 7.2% in >60 years of age group. In addition, the data represented predominance of *S. flexneri* infection (6.9%) in children aged between 6–14 years (Fig 3). *S. sonnei* infection dominates in the 6–14-year-old age group (1.9%), followed by <5 years of age (1.8%) (Fig 3). *S. flexneri* and *S. sonnei* isolation rates among the 0–5 year age group remained almost similar. Although there was no significant relation with respect to different serotype distribution among the other age groups (*p = 0.873*). However, the isolation rate of *S. sonnei* among the different age groups showed a significant variation (*p = 0.0001*). The isolation percentage among the 0–14 group was much higher than the other age groups. The S. *flexneri* isolation rate was significantly higher in age groups other than 0–5 years (*p = 0.0001*). In the case of *Shigella spp.*, the isolation rate among males was slightly higher (56.8%) than that of female patients (43.2%).

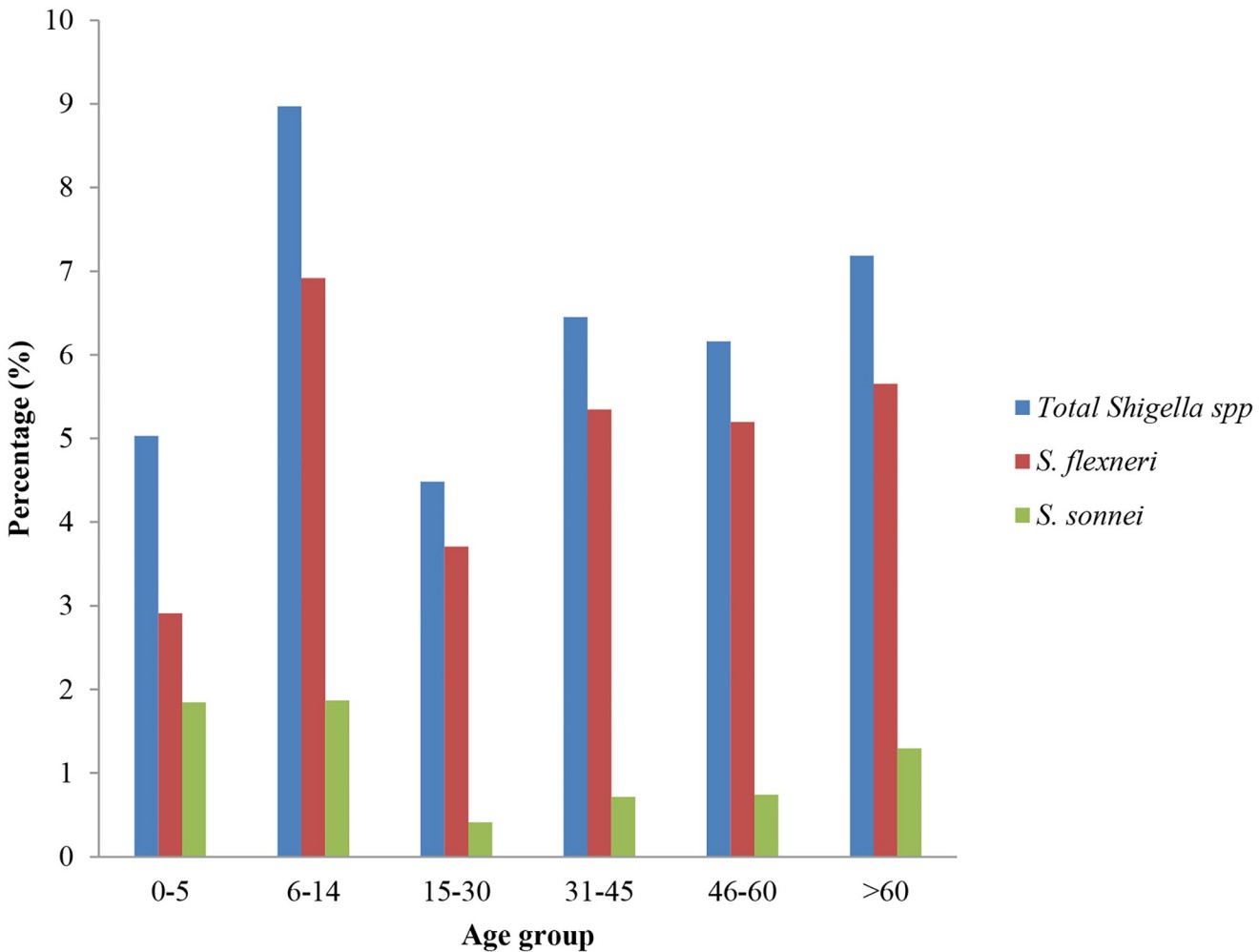

**Fig 3. Demographic features of *Shigella* cases.** Percentage (%): the proportion of *Shigella* cases by different groups of ages.

During the entire duration of the study, *S. flexneri* was the most predominant serogroup, among which serotype 2a (54%) had the highest rate, followed by serotypes 3a, 4a, and 1b, with prevalence rates of 21%, 8%, and 5%, respectively (Fig 4). Notably, 13 (4.2%) *S. flexneri* were designated as *S. flexneri* untypable (UT), as they only agglutinated with polyvalent type B antiserum; further serotyping of these isolates could not be completed with commercially available *Shigella* antisera.

## 2. Antimicrobial resistance in *Shigella* spp

A significant portion of isolates exhibited resistance to nalidixic acid (91.8%), ciprofloxacin (83.4%), ofloxacin (81.9%), norfloxacin (79.8%), tetracycline (75.5%), co-trimoxazole (65.6%), and streptomycin (69.9%). For chloramphenicol and ampicillin, the *Shigella* isolates showed a medium range of resistance (46.3% and 54.4%, respectively). For the third generation cephalosporin, and ceftriaxone, the susceptibility rate of the isolates was about 94%. All the isolates remained susceptible to the carbapenem group of antibiotics (Table 2). In *S. dysenteriae*, nalidixic acid and streptomycin resistance were high (80% and 70%, respectively). However, all *S. dysenteriae* isolates were susceptible to ceftriaxone.

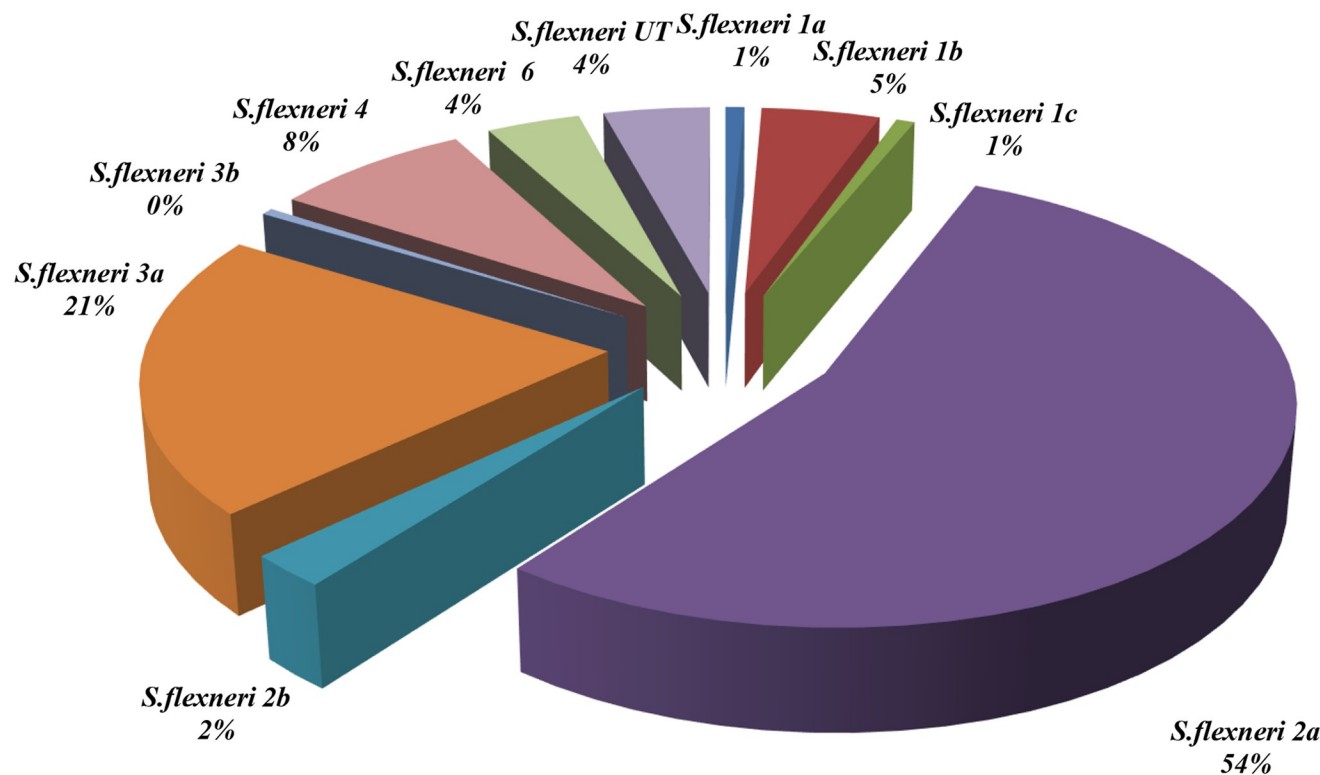

**Fig 4. The serotypes of *S. flexneri* species from 2011 to 2019 in Kolkata, India.**

Almost 80% of *S. flexneri* had multidrug resistance, whereas only 6.1% were resistant to ceftriaxone. *S. boydii* was susceptible to most of the tested drugs except streptomycin, nalidixic acid, ampicillin, and co-trimoxazole, with resistances of 94.1%, 64.7%, 64.7%, and 58.8%, respectively. On the other hand, *S. sonnei* isolates showed 70–90% resistance to all tested antimicrobials, except ampicillin, chloramphenicol, and ceftriaxone (13%, 4%, and 6%, respectively).

During this 9-year study period, the resistance rate of *S. flexneri* to ampicillin and ceftriaxone increased from 47.9% to 69.2% and 3.2% to 9.6%, respectively, whereas that of *S. sonnei*

**Table 2. The percentages of resistance among different *Shigella* serotypes to various antimicrobial agents.**

| Antimicrobial agents | Number resistant to antimicrobials (%) | | | | |
|---|---|---|---|---|---|
|  | Total (*n* = 535) | *S. dysenteriae* (*n* = 10) | *S. flexneri* (*n* = 407) | *S. boydii* (*n* = 17) | *S. sonnei* (*n* = 100) |
| Nalidixic Acid | 491(91.77) | 8(80) | 373(91.64) | 11(64.70) | 99(99) |
| Ciprofloxacin | 446(83.36) | 2(20) | 342(84.02) | 3(17.65) | 99(99) |
| Norfloxacin | 427(79.81) | 1(10) | 331(81.33) | 3(17.65) | 92(92) |
| Ofloxacin | 438(81.87) | 1(10) | 340(83.54) | 2(11.76) | 95(95) |
| Tetracycline | 404(75.51) | 5(50) | 329(80.84) | 3(17.65) | 67(67) |
| Ampicillin | 291(54.39) | 5(50) | 262(64.37) | 11(64.70) | 13(13) |
| Ceftriaxone | 32(5.98) | 0(0) | 25(6.14) | 1(5.88) | 6(6) |
| Co-trimoxazole | 351(65.60) | 5(50) | 259(63.63) | 10(58.82) | 77(77) |
| Streptomycin | 374(69.90) | 7(70) | 271(66.58) | 16(94.12) | 80(80) |
| Chloramphenicol | 248(46.35) | 3(30) | 240(58.97) | 1(5.88) | 4(4) |

**Table 3. Minimum inhibitory concentrations MIC of antimicrobial agents for *Shigella* spp.**

| Antimicrobial agent | Break point (µg/ml) | Range | *S. dysenteriae* | | *S. flexneri* | | *S. boydii* | | *S. sonnei* | |
|---|---|---|---|---|---|---|---|---|---|---|
| | | | MIC$_{50}$ (µg/ml) | MIC$_{90}$ (µg/ml) | MIC$_{50}$ (µg/ml) | MIC$_{90}$ (µg/ml) | MIC$_{50}$ (µg/ml) | MIC$_{90}$ (µg/ml) | MIC$_{50}$ (µg/ml) | MIC$_{90}$ (µg/ml) |
| Ampicillin | ≥32 | 32–512 | >512 | >512 | >256-<512 | >512-<1024 | >128-<256 | >512 | >32-<64 | >32-<64 |
| Tetracycline | ≥16 | 16–256 | >64-<128 | >64-<128 | >64-<128 | >64-<128 | >16-<32 | >32-<64 | >64-<128 | >64-<128 |
| Ciprofloxacin | ≥4 | 4–64 | >4-<8 | >4-<8 | >4-<8 | >16-<32 | <4 | <4 | >4-<8 | >8-<16 |
| Nalidixic Acid | ≥32 | 32–1024 | >128-<256 | >128-<256 | >1024 | >1024 | >32-<64 | >256-<512 | >512-<1024 | >512-<1024 |
| Streptomycin | - | 64–1024 | >64-<128 | >64-<128 | >512-<1024 | >1024 | >64-<128 | >64-<128 | >256-<512 | >1024 |
| Chloramphenicol | ≥32 | 32–512 | <32 | <32 | >32-<64 | >64-<128 | <32 | >32-<64 | >16-<32 | >32-<64 |

increased from 0% to 17.1% and 0% to 6.9%, respectively. There was a notable increase in resistance to ciprofloxacin ($p = 0.000$), ofloxacin ($p = 0.058$), ampicillin ($p = 0.0088$) and ceftriaxone ($p = 0.0106$). There is a decreasing trend of resistance to co-trimoxazole. While there was no significant change in the *S. sonnei* AMR trend, *S. flexneri* isolates showed significant changes in the resistance for ampicillin, co-trimoxazole, and ceftriaxone ($p = 0.0001$, $0.0004$, and $0.0105$, respectively).

The MIC range for different serotypes is shown in Table 3. Several types of resistance profiles were observed, and the common resistance profiles have been depicted in Table 4. The majority of the *Shigella* isolates demonstrated resistance to multiple categories of antibiotics. The most common resistance profile of the isolates confered resistance to almost five groups of antibiotics, like aminoglycosides, tetracycline, antifolate, fluoroquinolones, and β-lactam.

## 3. Detection of virulence genes

About 51% of the isolates harboured *ial*, which was encoded in an invasive plasmid. *virF* and *virB* were found in 50.9% and 40.8% of the isolates, respectively. The chromosomally encoded *set* was present in 39.5% of the tested isolates, whereas the plasmid-encoded *sen* was found in 48.4% of the isolates. The secreted autotransporter toxin-encoded *sat* gene was found in 52.2% of the isolates, and other serine protease autotransporter toxin encoding genes like *sigA*, *pic*, and *sepA* were present in 63%, 35.7%, and 53.5% of the isolates, respectively.

## 4. Characterisation of integrons and AMR genes

A total of 28.6%, 26.12%, and 40.13% of the *Shigella* isolates harboured *intlI*, *intlII*, and both *intlI/intlII* genes, respectively. No class *intIII* was noted in any of the *Shigella* isolates. The prevalence of *intI*, *intII* and both *intI/intII* genes in different serogroups of *Shigella* spp. is shown in Fig 5.

**Table 4. Most common Resistance profile of *Shigella* isolates of ID hospital during 2011–2019.**

| Sl No | Resistance profile | Total no. of Isolates | 2011 | 2012 | 2013 | 2014 | 2015 | 2016 | 2017 | 2018 | 2019 |
|---|---|---|---|---|---|---|---|---|---|---|---|
| I | NA/CIP/NOR/OFX/TET/AM/SXT/S/CHL | 100 | 12 | 22 | 3 | 7 | 11 | 12 | 12 | 10 | 11 |
| II | NA/CIP/NOR/OFX/TET/SXT/S | 70 | 5 | 11 | 2 | 7 | 13 | 9 | 14 | 5 | 4 |
| III | NA/CIP/NOR/OFX/TET/AM/S/CHL | 57 | 0 | 4 | 0 | 6 | 5 | 3 | 21 | 11 | 7 |
| IV | NA/CIP/NOR/OFX/TET/SXT/S/CHL | 38 | 8 | 10 | 3 | 2 | 1 | 1 | 7 | 2 | 4 |
| V | NA/CIP/NOR/OFX/TET/AM/SXT/S | 22 | 5 | 0 | 0 | 1 | 7 | 1 | 2 | 2 | 4 |
| VI | NA/CIP/NOR/OFX | 17 | 1 | 2 | 1 | 1 | 3 | 2 | 3 | 2 | 2 |
| VII | NA/CIP/NOR/OFX/S | 14 | 0 | 1 | 1 | 0 | 0 | 3 | 5 | 1 | 3 |

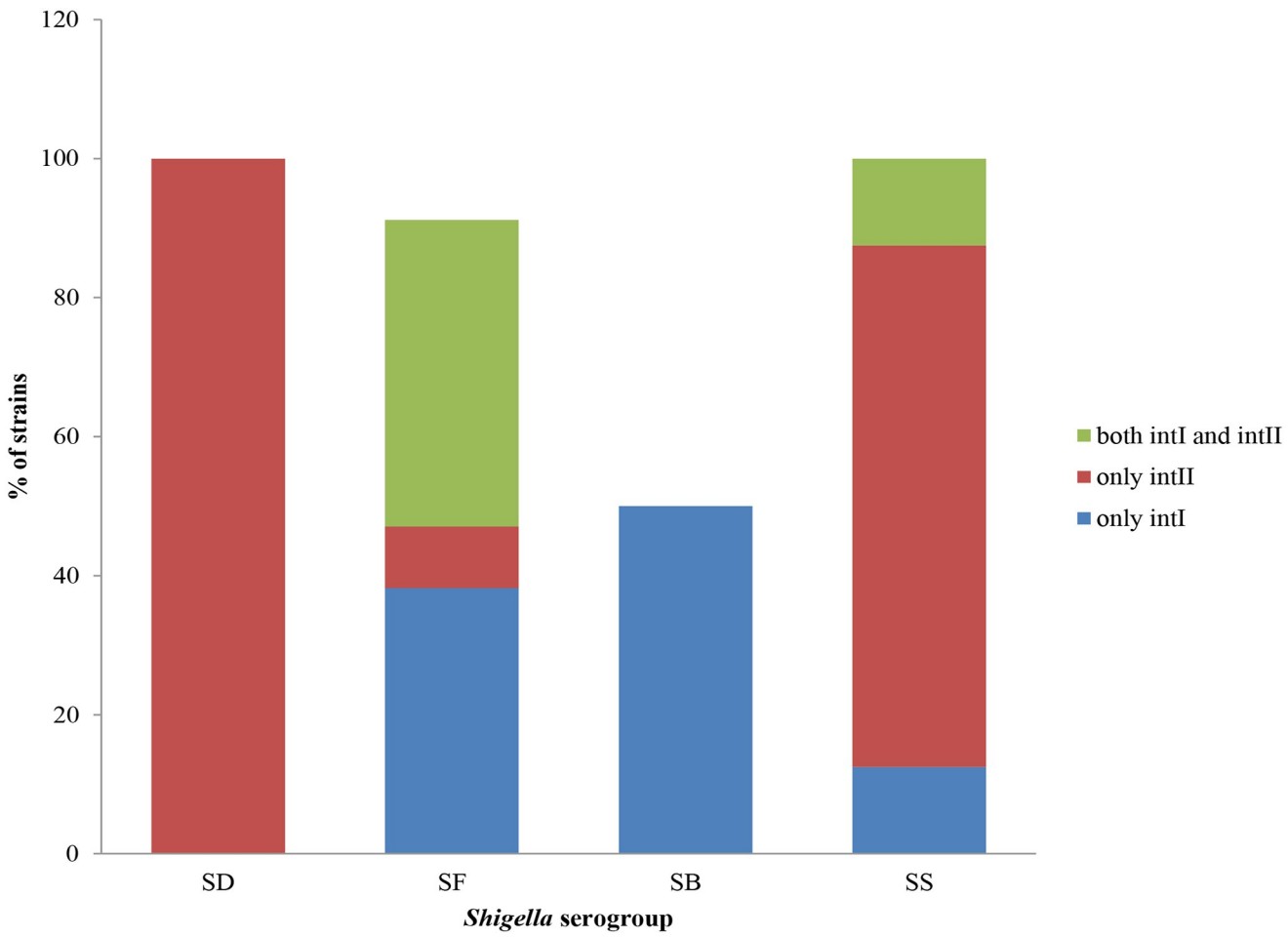

**Fig 5. Distribution of different intrgrons in different serogroups of *Shigella*.**

The presence of atypical class 1integrons and typical class 1 integrons was detected in 17.8% and 5% of the *Shigella* isolates, respectively. While 17.4% of the isolates were found to carry both class 1 typical and class 1 atypical integrons. Interestingly, different types of class 1 integrons co-existed with class 2 integrons, with 16.6% of the isolates carrying the combination of atypical class 1/class 2 integrons, while only 7% of the isolates had typical class 1/class 2 integrons (Fig 6).

The strains with typical integrons showed two different sizes of amplicons (500 bp and >2.5 kb). On sequencing, the 500 bp of the class 1 integron did not show any of the antimicrobial resistance gene cassettes. Atypical class 1 integrons had 2 kb and 2.4 kb amplicon sizes, and upon sequencing, the 2kb had 100% nucleotide identity with GenBank accession no. GQ214137 of *S. flexneri* with the AMR gene cassettes *dfrA1-aadA*, and *bla*$_{OXA}$-*aadA* gene cassettes with full sequence similarity with GenBank accession no. KX817769 was carried by 2.4 kb variable region of the atypical class 1 integron. Two different sizes of class 2 gene cassettes were found among the tested isolates. The 1.4 kb cassette was observed mainly in *S. sonnei*, whereas the 2.2 kb region was found in other *Shigella* spp. The former had a sequence similarity *dfrA1-sat* (GenBank accession no. KX817766), and the later shared the *dfrA1-sat-aadA* sequence similarity (GenBank accession no. KX817767).

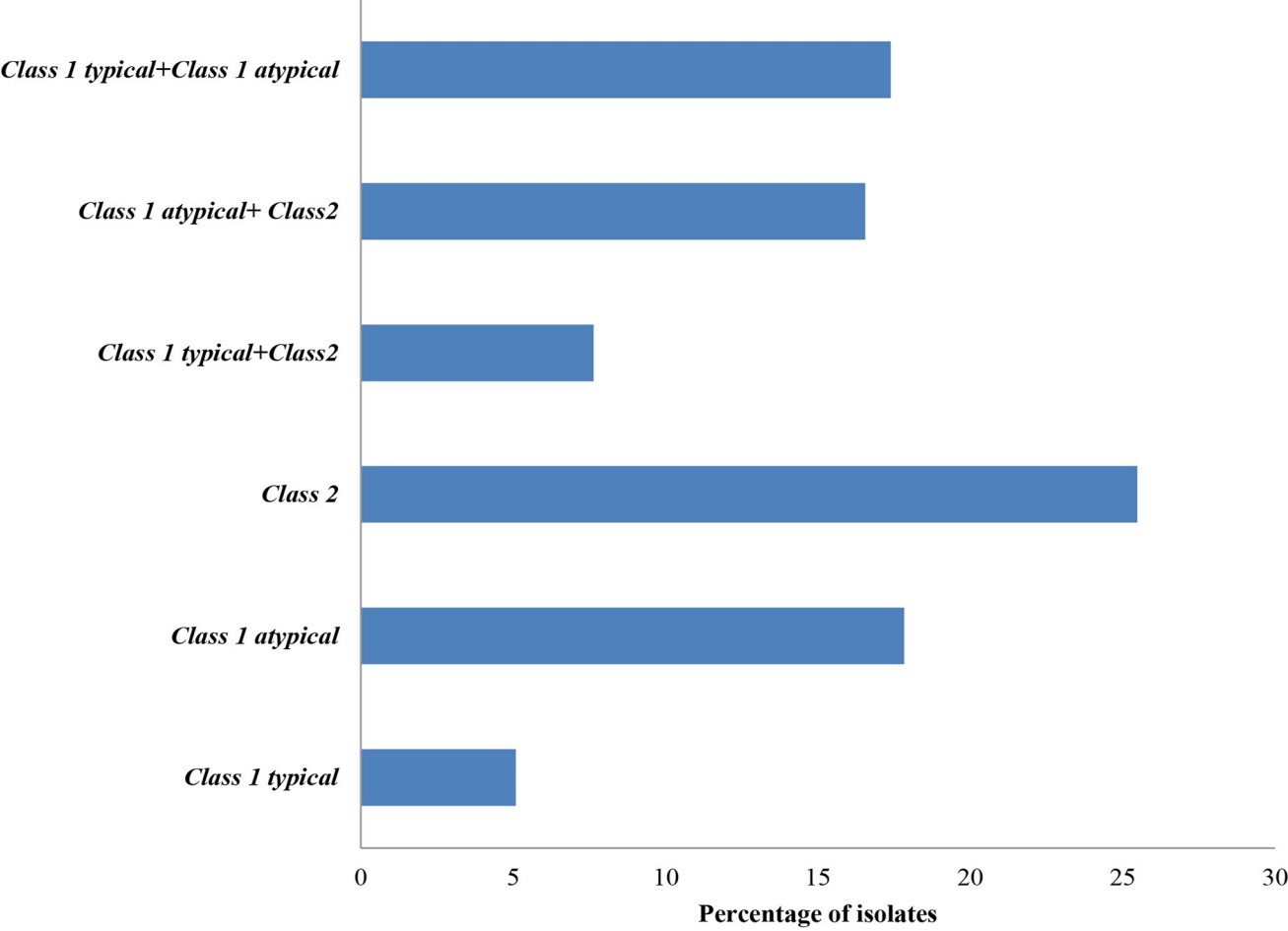

**Fig 6. Distribution of different classes of integrons among tested isolates.**

The presence of AMR genes was screened among isolates that were resistant to one or more antimicrobials. Among these, 69.4% were positive for $bla_{OXA-1}$, whereas only 12.3% were positive for $bla_{TEM-9}$. Whereas, among 32 ceftriaxone resistance strains 37.5% were $bla_{CTX-M}$ positive. *tetA* and *tetB* genes were respectively detected in 13.9% and 77.9% of the isolates that confer resistance to tetracycline. *qnrB* (4.6%), *qnrS* (3.8%),and *aac (6')1b* (0.8%) were found to be responsible for fluoroquinolone resistance. Mostly, quinolone resistance was associated with *gyrA* (S83L, D87N, H211Y) and/or *parC* (S80I) mutations (Table 5). No mutation was seen in the *gyrB* and *parE*. Other AMR-encoding genes like *strA* and *aadA1A* for streptomycin and *dfrA1* and *sul2* for co-trimoxazole were detected in 52.4%, 55.8%, 84%, and 61.9%, respectively, whereas the prevalence of the *catA*, responsible for chloramphenicol resistance was found 46.1% of the isolates.

## 5. Plasmid profiling

The dendrogram-based genetic similarity analysis of representative *Shigella* spp. was analysed. There was a similarity of 77.6% among the *S. boydii* group (S3 Fig), whereas the similarities were less among *S. flexneri* and *S. sonnei* (18% and 25.6%, respectively) (S4 and S5 Figs), indicating the diversity of plasmid patterns.

**Table 5. Mutations in quinolone resistance determining regions (QRDR) of representative quinolone resistant *Shigella* isolates.**

| Strain | Strain ID | Quinolone resistance | MIC CIP(µg/ml) | *GyrA* | *ParC* |
|---|---|---|---|---|---|
| *S. flexneri 2a* | IDH10994 | NA, CIP, NOR, OFX | 4–8 | S83->L<br>D87->N<br>H211->Y | S80->I |
| *S. sonnei* | IDH 8898 | NA, CIP, NOR, OFX | 8–16 | S83->L<br>D87->N | S80->I |
| *S. flexneri 2a* | IDH 10932 | NA, CIP, NOR, OFX | 16–32 | S83->L<br>D87->N<br>H211->Y | S80->I |
| *S. dysenteriae* | IDH 9602 | NA, CIP, NOR, OFX | 2–4 | S83->L | - |
| *S. sonnei* | IDH 6701 | NA, CIP, NOR, OFX | 0.5–1 | S83->L<br>D87->G | S80->I |

## 6. Molecular typing by PFGE

The *Xba*I DNA fragments obtained in the PFGE were analysed with the corresponding species: *S. dysenteriae* (Fig 7), *S. flexneri* (Fig 8), *S. boydii* (Fig 9) and *S. sonnei* (Fig 10). PFGE analysis of *S. dysenteriae* isolates revealed two clusters with approximately 70% similarity within them (Fig 7). When *S. flexneri* 2a isolates of different years and AMR profiles were compared, there were two clusters present with 85% similarity (Fig 8). The IDH 12673 isolate was identified as belonging to a different clone with only 74% similarity to other *S. flexneri* 2a. Among the 9 *S.*

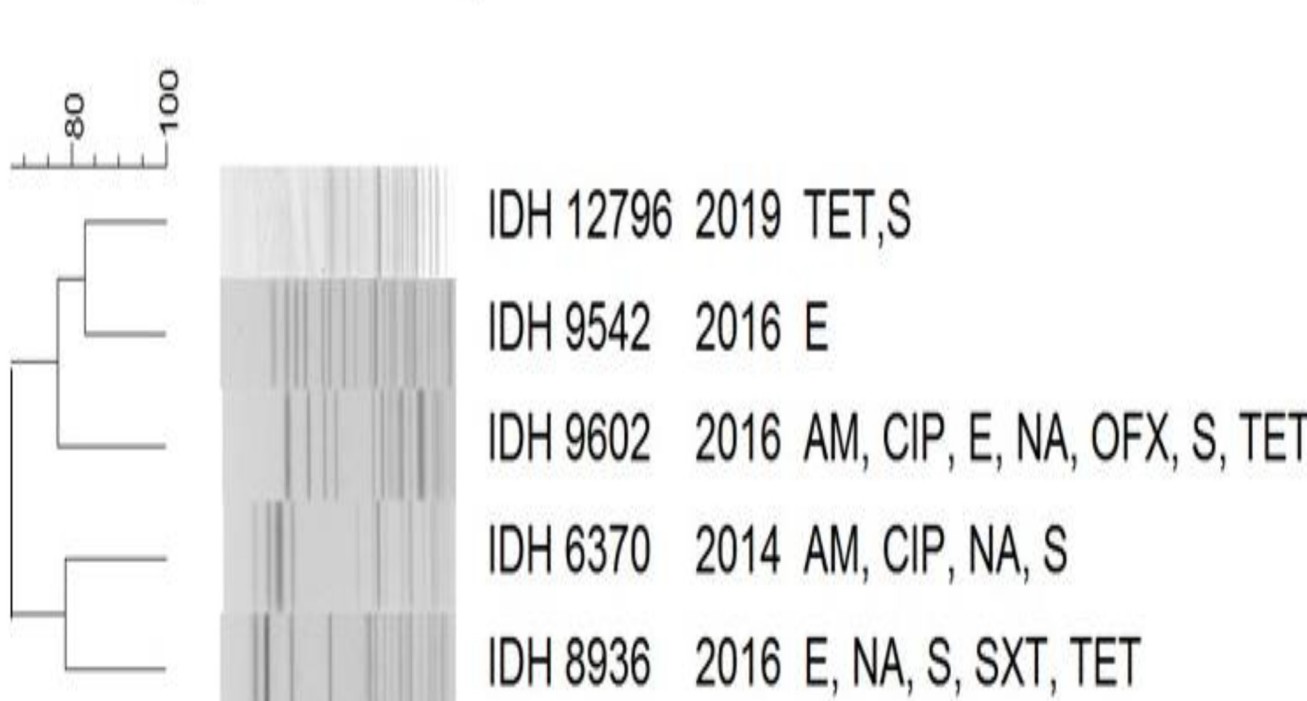

**Fig 7. *Xba*I-PFGE profiles of with dendrogram showing percentage similarity of *S. dysenteriae* isolates.**

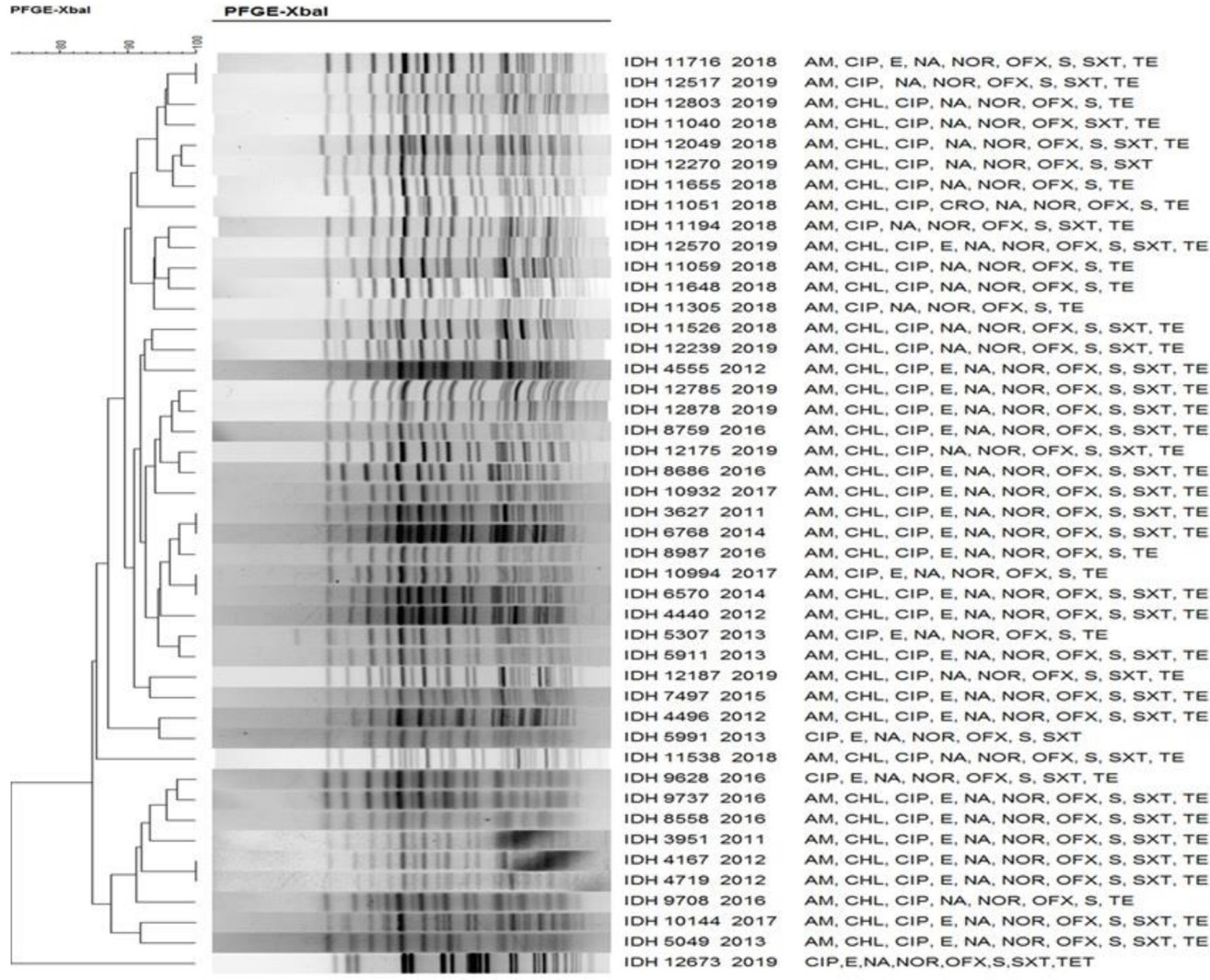

**Fig 8.** *Xba*I-PFGE profiles of with dendrogram showing percentage similarity of *S. flexneri* isolates.

*boydii* isolates, 6 strains formed a distinct cluster with 76% similarity to another cluster of 3 isolates (Fig 9). *S. sonnei* had two clusters with 85% similarity. Among 32 representative isolates, 18 belonged to the first cluster, whereas another 14 strains belonged to the second cluster and shared 90–100% similarity among the isolates (Fig 10). Only IDH 7727 shows a unique banding pattern with 83% similarity to the other isolates.

## Discussion

The frequency of *Shigella* isolation, the species distribution, and the antimicrobial resistance pattern were examined in this study. The total isolation rate was 5.8%, with *S. flexneri* (76.1%) being the most dominant one, followed by *S. sonnei* (18.7%), *S. boydii* (3.4%), and *S. dysenteriae* (1.8%). A similar trend has been reported from Bangladesh and China [27,28]. Our finding indicates that the overall isolation rate of *Shigella* spp. from Kolkata remains 6–7% and the prevalence of its species remained the same as reported before from the same region [23,29].

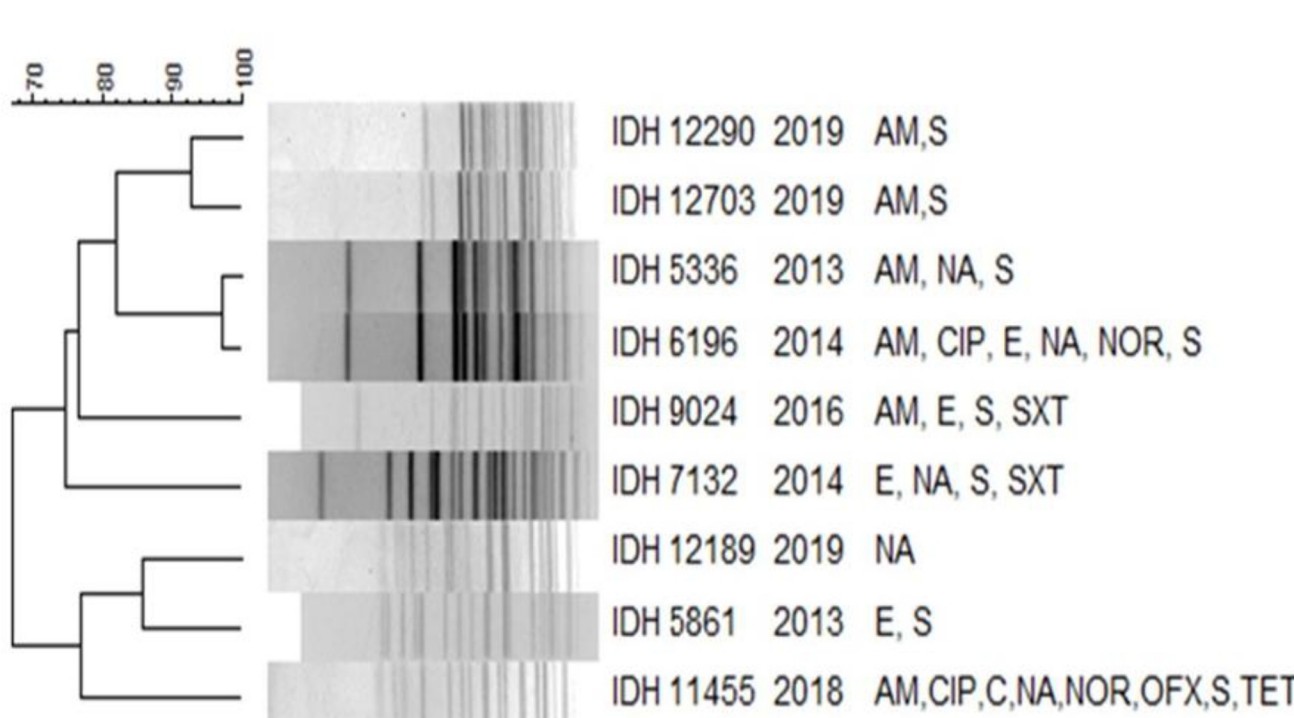

**Fig 9. *Xba*I-PFGE profiles of with dendrogram showing percentage similarity of *S. boydii* isolates.**

Two investigations conducted in Kolkata between 2001 and 2007 [30,31] and one conducted in North Karnataka over a period of 12 years [32] found that the frequency of *S. dysenteriae* was 2.1% greater than that of *S. boydii*, which slightly differs from our findings. Numerous investigations around the world [33–38] revealed that the most common serotype was *S. flexneri* 2a, which was also observed in this study. We found that the children belonging to the age group 6–14 years had the largest proportion of being infected with *S. flexneri* or *S. sonnei*. A study conducted in Asia and Africa indicated that children aged below 5 years are mostly infected with *Shigella* spp. [39,40]. One plausible explanation for the lack of cases in children younger than 5 years old in the current study might be due to the fact that children with mild to moderate diarrhea would have received treatment at home or in other hospitals in the city. Like in other reports [41,42], the ratio of male to female *Shigella*-infected cases is similar.

*Shigella* pathogenesis is the result of a combination of several different virulence factors. Almost all the species of *Shigella* had plasmid-borne *inv* and also the chromosomally encoded virulence factors.

In 2017, the WHO highlighted the urgent need for new antibiotics on a list of priority pathogens, and *Shigella* was among them. More than 60% of the isolates in our investigation were resistant to fluoroquinolone, aminoglycoside, and folate inhibitory groups, which matched with other global reports [20,22,43–45]. However, *S. sonnei* was more sensitive to ampicillin and chloramphenicol than *S. flexneri*. Ceftriaxone and azithromycin are currently recommended by the WHO for the treatment of fluoroquinolone-resistant *Shigella* infections. Our

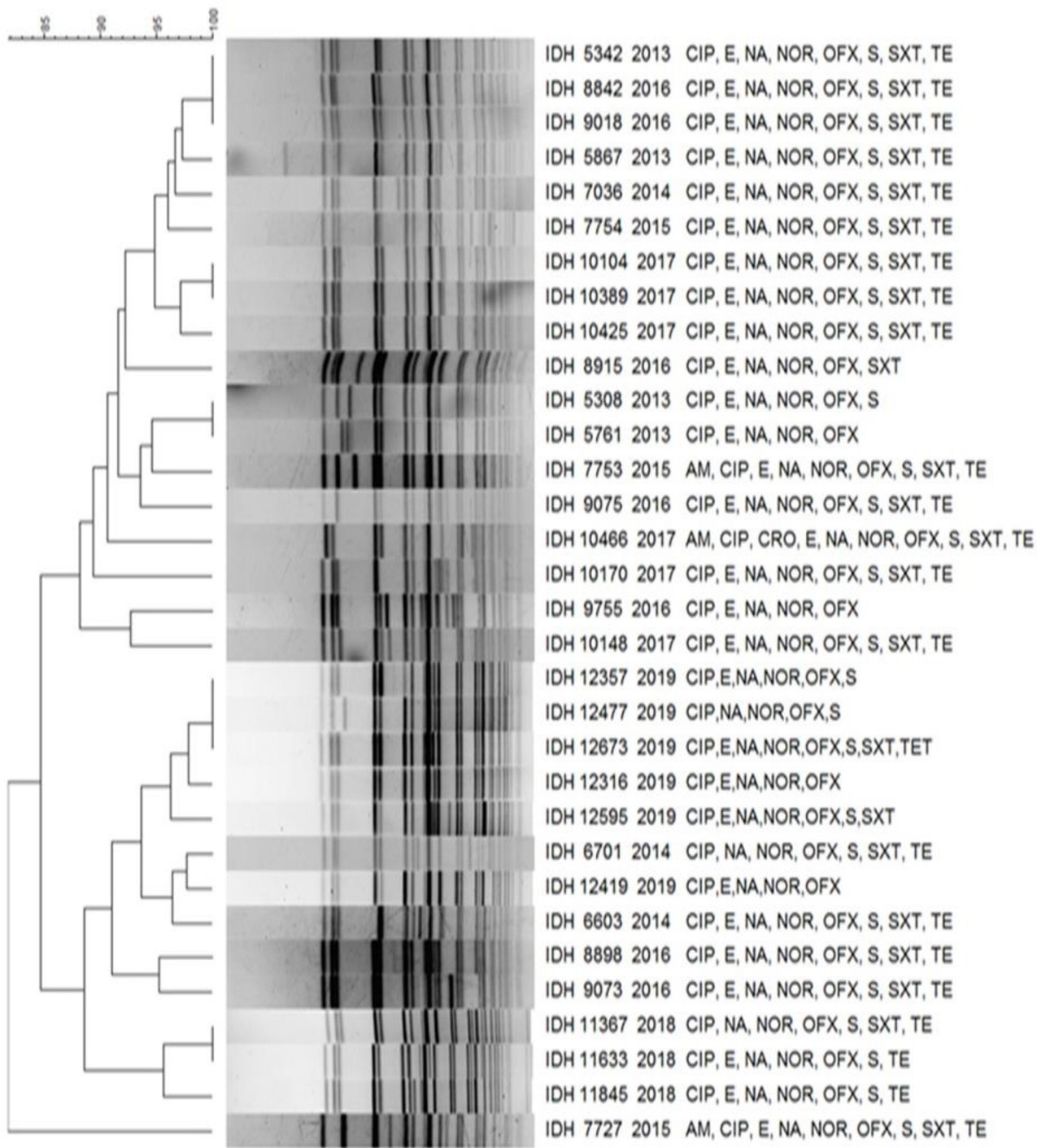

**Fig 10. *Xba*I-PFGE profiles of with dendrogram showing percentage similarity of *S. sonnei* isolates.**

finding indicates that *Shigella* was highly susceptible to cephalosporin, suggesting that this drug could be a better choice for empirical treatment.

According to a CDC's report, approximately 77,000 drug-resistant *Shigella* infections occur annually, making this pathogen categorized as a serious threat [46]. Determining the role of genetic elements such as integrons, plasmids, and chromosomally carried topoisomerase mutations in the development of highly multi-drug-resistant phenotypes is therefore crucial. Certain *Shigella* isolates have developed resistance to antibiotics like trimethoprim and spectinomycin because of integrons [47,48]. Previous studies have indicated that a majority of the isolates examined had an atypical class 1 integron, but that a few had the typical class 1 integron [49]. In class two integrons, *dfrA1-sat* and *dfrA1-sat-aadA* are the most common gene cassettes identified in *S. sonnei* and *S. flexneri*, respectively. Class 2 integrons seem to be the distinctive feature of these two *Shigella* serogroups.

The mobile genetic elements, including integrons and plasmids are playing a significant role in the spread of AMR among bacteria. These bacteria frequently have $bla_{TEM}$, which imparts resistance to penicillin and other β-lactam antibiotics [50,51]. Nevertheless, the OXA-type β-lactam, with significant hydrolytic activity against oxacillin and cloxacillin, frequently transmits resistance to ampicillin and cephalothin. Similar to the other studies, our findings show that 69.4% of the isolates tested positive for $bla_{OXA-1}$ and 12.3% for $bla_{TEM}$ [21,52]. In accordance with earlier results [52], *tet(B)* was more frequent than *tet(A)*. Streptomycin resistance was largely attributed to *strA*, but *aadA1* also appears to play a strong role. Quinolone resistance is typically caused by mutations in QRDR region, efflux pump activity, and PMQR genes [53]. In this study, several mutations in the QRDR have been identified, irrespective of MIC. Mutations in the *gyrA* (S83, D87, H211) and *parC* (S80) have been frequently reported from India as well as globally [54]. Although no mutation was noted in *gyrB* gene in our study whereas only one single mutation at 517 position is reported from China [54]. An association of broader range of resistance to different quinolone was also observed with mutations in both topoisomerases [55].

Our study has identified the presence of the PMQR genes *qnrB*, *qnrS*, and *aac(6')1b* that are responsible for fluoroquinolone resistance. Empirical treatments rely heavily on antibiotic resistance patterns since it can also be used as a typing system and AMR determinants spreading indicator as well [19]. Due to the absence of diversity in the susceptibility patterns, we were unable to use the AMR pattern as an epidemiological marker.

Molecular epidemiological investigations, comparing plasmid and PFGE profiles, are valuable tools for determining the probable relatedness among clinical isolates of a specific bacterial species [56–58]. Our study ascertained that the clonal relatedness among *S. dysenteriae* isolates was 70%, which is almost similar to previous reports from Kolkata [23,49,59]. In *S. flexneri*, 85% similarity was noted, which is also similar to other findings [23,59–62]. As shown in a previous study from Kolkata, representative isolates of *S. boydii* had 76% similarity [49]. The clonal relatedness of *S. sonnei* from different countries reproduces a parallel observation like our study outcome [21,27,52,62–64]. As evidenced by this study, multiple clones were identified in different serotypes, confirming the existence of diverse origins among *Shigella* spp.

## Limitations

There are a number of constraints that have been identified during the nine-year monitoring period. Local administration, individuals at risk, government agencies, healthcare providers at all levels, and healthcare providers at the federal and state levels must all work together continuously to maintain comprehensive sentinel surveillance. Since this bacterium can be

transmitted from person to person through the faecal-oral route and can be found in both water and food, it has been extremely difficult to persuade people of the importance of public health. Further complications such as missed diagnoses, drug resistance, and limited treatment options result from the unsupervised use of antibiotics for diarrhea. One possible explanation for the fluctuating reporting of cases over time is that healthcare workers and other vulnerable populations are not consistently informed about this bacterium. The potential underreporting of subclinical and distant cases is a noteworthy worry associated with this surveillance. Strains prevalent in rural areas are not included in the study because it only includes cases admitted to the ID hospital as referral patients. The study sheds light on the most prevalent strains in Kolkata, but it doesn't go far enough to compare their clonality to other parts of the Indian subcontinent.

## Conclusion

Due to overcrowding and poor sanitation, shigellosis is mainly found in developing nations. The species *S. flexneri* and *S. sonnei* were found most frequently, with *S. dysenteriae* being an exception. According to the World Health Organization's guidelines (2016), the risk of death from Shigellosis is higher in infants, nonbreastfed children, children recovering from measles, malnourished children, and adults older than 50 years. The two main *Shigella* species, according to our results, caused shigellosis in different age groups. Additionally, this study helps with virulence profiling of *Shigella* strains and serotype distribution profiling, both of which are important for tracking new variants and patterns of isolates in our region; many questions remain about this notorious and evasive bacterial pathogen. As the research progressed, resistance to cotrimoxazole decreased while resistance to anti-shigellosis medications increased. There has been a shift in the antibiotic resistance pattern and the frequency of certain *Shigella* species at various points in time. In order to select effective medications for empirical treatment, it is essential to thoroughly investigate the antibiotic susceptibility pattern and the precise geographical source of antibiotic resistance. An enormous problem could arise from the unchecked use of antibiotics in general, since this would hasten the development of multi-resistant bacteria that pose a threat to current treatment methods. Therefore, it is essential to learn how resistance works and devise methods for eliminating it. Public health measures, such as clean water and sanitation, as well as public health education, are necessary to reduce the morbidity and mortality caused by diarrhea in our nation. Other important factors include newer antimicrobials and cost-effective vaccines. In order to better understand local circulating strains and their phenotypic and genotypic traits, it is recommended that mandatory surveillance be expanded to include shigellosis in both hospitals and the community. Together, these initiatives have the potential to stem the tide of antimicrobial-resistant *Shigella* from spreading over the world. We conclude that antimicrobial treatment should be revised on a regular basis according to surveillance data.

## Supporting information

**S1 Table. Primer list for virulence gene.**
(DOCX)

**S2 Table. Primer list for antimicrobial resistance gene.**
(DOCX)

**S1 Fig. Trends for *S. sonnei* isolation rate with relation to seasonality.**
(TIF)

**S2 Fig. The trends for *S.flexneri* isolation rate with relation to seasonality.**
(TIF)

**S3 Fig. Similarity among the *Shigella boydii*isolateson the basis of their plasmid profiles.**
(TIF)

**S4 Fig. Similarity among the *Shigella flexneri 2a* isolates on the basis of their plasmid profiles.**
(TIF)

**S5 Fig. Similarity among the *Shigella sonnei* isolates on the basis of their plasmid profiles.**
(TIF)

## Acknowledgments

We are thankful to Mr. Mukul Roy and Mr. Biswajit Sharma for technical assistance in the PFGE experiment.

## Author Contributions

**Conceptualization:** Puja Bose, Shanta Dutta, Asish Kumar Mukhopadhyay.

**Data curation:** Puja Bose, Gourab Halder.

**Formal analysis:** Puja Bose, Goutam Chowdhury, Gourab Halder, Alok K. Deb.

**Funding acquisition:** Kei Kitahara, Shin-ichi Miyoshi, Masatomo Morita.

**Investigation:** Puja Bose, Gourab Halder, Asish Kumar Mukhopadhyay.

**Methodology:** Puja Bose, Gourab Halder, Debjani Ghosh.

**Project administration:** Kei Kitahara, Thandavarayan Ramamurthy, Shanta Dutta, Asish Kumar Mukhopadhyay.

**Resources:** Shin-ichi Miyoshi, Shanta Dutta, Asish Kumar Mukhopadhyay.

**Software:** Goutam Chowdhury, Gourab Halder, Alok K. Deb, Masatomo Morita.

**Supervision:** Shanta Dutta, Asish Kumar Mukhopadhyay.

**Validation:** Thandavarayan Ramamurthy, Asish Kumar Mukhopadhyay.

**Visualization:** Puja Bose, Asish Kumar Mukhopadhyay.

**Writing – original draft:** Puja Bose, Gourab Halder.

**Writing – review & editing:** Goutam Chowdhury, Gourab Halder, Thandavarayan Ramamurthy, Asish Kumar Mukhopadhyay.

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
