## [Decision Letter · Decision Letter 0]

27 Dec 2023

Dear Author,

Thank you very much for submitting your manuscript "Prevalence and changing antimicrobial resistance profiles of Shigella spp. isolated from diarrheal patients in Kolkata during 2011-2019" for consideration at PLOS Neglected Tropical Diseases. As with all papers reviewed by the journal, your manuscript was reviewed by members of the editorial board and by several independent reviewers. The reviewers appreciated the attention to an important topic. Based on the reviews, we are likely to accept this manuscript for publication, providing that you modify the manuscript according to the review recommendations. 

Dear Author, 

The manuscript requires minor modification as suggested below.

Reviewer 1:- No changes suggested

The study is well planned and all methods followed are standard. This is a retrospective analysis, they have achieved all the objectives. The results are in line with the analysis plan with all the results well depicted and clearly stated. The conclusions are supported by the data presented. The authors have addressed public helath as well as importance of antibioc resistance surveillance emphasizing development of treatment guidelines. This is a retrospective analysis which is well planned. The study highlights the importance of evolving antibiotic resistance with clonal diversity.

Reviewer 2:- Minor Revision

The authors retrospectively analysed the occurrence, characteristics, and antimicrobial resistance patterns of various Shigella serotypes isolated from patients with acute diarrhea of the Infectious Diseases Hospital in Kolkata from 2011–2019.

The following are the minor comments to be included by the authors:

Study design, statistical analysis method and ethical approval to be mentioned.

The conclusion and limitation of the study, if any to be mentioned. 

1. Line 110: Mention the references of studies that have been made across the world to track the epidemiology and antimicrobial susceptibility patterns of Shigella spp.

2. Mention the stool macroscropy characteristics (bloody diarrhoea/ watery diarrhoea)

3. Whether all Shigella spp. were typable or few non-typable strains were there.

4. Please mention why during AMR gene detection for β-lactamase genes blaOXA and blaTEM was choosed and not CTX-M?

5. Mention in discussion regarding the gryB and parC prevalence in comparison to your findings both in India and globally

6. Grammatical corrections needed in the introduction and discussion part.

7. Conclusion and limitation of the study, if any to be mentioned.

8. Study design, statistical analysis method and ethical approval to be mentioned.

9. Ethical approval to be mentioned.

Reviewer 3:- A few minor edits are needed, as indicated in the reviewed file.

The study has been carried out systematically and study design and analysis have been done appropriately. The results have been delineated very systemically and presented appropriately. The study highlights the importance emerging resistance and emphasizes the evolving resistance pattern over a nine-year period from 2011 to 2019. The authors have carried out the study systematically, clearly delineated the findings from the testing and emphasize the emerging resistance patterns. 

Regards,

Sincerely,

Sivanantham Krishnamoorthi

Guest Editor

Stuart Blacksell

Section Editor

Dear Author, 

The manuscript requires minor modification as suggested below.

Reviewer 1:- No changes suggested

The study is well planned and all methods followed are standard. This is a retrospective analysis, they have achieved all the objectives. The results are in line with the analysis plan with all the results well depicted and clearly stated. The conclusions are supported by the data presented. The authors have addressed public helath as well as importance of antibioc resistance surveillance emphasizing development of treatment guidelines. This is a retrospective analysis which is well planned. The study highlights the importance of evolving antibiotic resistance with clonal diversity.

Reviewer 2:- Minor Revision

The authors retrospectively analysed the occurrence, characteristics, and antimicrobial resistance patterns of various Shigella serotypes isolated from patients with acute diarrhea of the Infectious Diseases Hospital in Kolkata from 2011–2019.

The following are the minor comments to be included by the authors:

Study design, statistical analysis method and ethical approval to be mentioned.

The conclusion and limitation of the study, if any to be mentioned. 

1. Line 110: Mention the references of studies that have been made across the world to track the epidemiology and antimicrobial susceptibility patterns of Shigella spp.

2. Mention the stool macroscropy characteristics (bloody diarrhoea/ watery diarrhoea)

3. Whether all Shigella spp. were typable or few non-typable strains were there.

4. Please mention why during AMR gene detection for β-lactamase genes blaOXA and blaTEM was choosed and not CTX-M?

5. Mention in discussion regarding the gryB and parC prevalence in comparison to your findings both in India and globally

6. Grammatical corrections needed in the introduction and discussion part.

7. Conclusion and limitation of the study, if any to be mentioned.

8. Study design, statistical analysis method and ethical approval to be mentioned.

9. Ethical approval to be mentioned.

Reviewer 3:- A few minor edits are needed, as indicated in the reviewed file.

The study has been carried out systematically and study design and analysis have been done appropriately. The results have been delineated very systemically and presented appropriately. The study highlights the importance emerging resistance and emphasizes the evolving resistance pattern over a nine-year period from 2011 to 2019. The authors have carried out the study systematically, clearly delineated the findings from the testing and emphasize the emerging resistance patterns. 

Regards,

Reviewer's Responses to Questions

**Key Review Criteria Required for Acceptance?**

**Methods**

-Are the objectives of the study clearly articulated with a clear testable hypothesis stated?

-Is the study design appropriate to address the stated objectives?

-Is the population clearly described and appropriate for the hypothesis being tested?

-Is the sample size sufficient to ensure adequate power to address the hypothesis being tested?

-Were correct statistical analysis used to support conclusions?

-Are there concerns about ethical or regulatory requirements being met?

Reviewer #1: The study is well planned and all methods followed are standard. This is a retrospective analysis, they have achieved all the objectives.

Reviewer #2: Study design, statistical analysis method and ethical approval to be mentioned

Reviewer #3: The study has been carried out systematically and study design and analysis have been done appropriately

**Results**

-Does the analysis presented match the analysis plan?

-Are the results clearly and completely presented?

-Are the figures (Tables, Images) of sufficient quality for clarity?

Reviewer #1: The results are in line with the analysis plan with all the results well depicted and clearly stated.

Reviewer #2: Yes

Reviewer #3: The results have been delineated very systemically and presented appropriately.

**Conclusions**

-Are the conclusions supported by the data presented?

-Are the limitations of analysis clearly described?

-Do the authors discuss how these data can be helpful to advance our understanding of the topic under study?

-Is public health relevance addressed?

Reviewer #1: The conclusions are supported by the data presented. The authors have addressed public helath as well as importance of antibioc resistance surveillance emphasizing development of treatment guidelines.

Reviewer #2: Conclusion and limitation of the study, if any to be mentioned

Reviewer #3: The study highlights the importance emerging resistance and emphasizes the evolving resistance pattern over a nine-year period from 2011 to 2019

**Editorial and Data Presentation Modifications?**

Reviewer #1: No changes suggested

Reviewer #2: Minor Revision

Reviewer #3: Table 1 can be reduced in size to fit in

**Summary and General Comments**

Reviewer #1: This is a retrospective analysis which is well planned. The study highlights the importance of evolving antibiotic resistance with clonal diversity.

Reviewer #2: The authors retrospectively analysed the occurrence, characteristics, and antimicrobial resistance patterns of various Shigella serotypes isolated from patients with acute diarrhea of the Infectious Diseases Hospital in Kolkata from 2011–2019.

The following are the minor comments to be included by the authors:

1. Line 110: Mention the references of studies that have been made across the world to track the epidemiology and antimicrobial susceptibility patterns of Shigella spp.

2. Mention the stool macroscropy characteristics (bloody diarrhoea/ watery diarrhoea)

3. Whether all Shigella spp. were typable or few non-typable strains were there.

4. Please mention why during AMR gene detection for β-lactamase genes blaOXA and blaTEM was choosed and not CTX-M?

5. Mention in discussion regarding the gryB and Par C prevalence in comparison to your findings both in India and globally

6. Grammatical corrections needed in the introduction and discussion part.

7. Conclusion and limitation of the study, if any to be mentioned.

8. Study design, statistical analysis method and ethical approval to be mentioned.

Reviewer #3: The authors have carried out the study systematically, clearly delineated the findings from the testing and emphasize the emerging resistance patterns

PLOS authors have the option to publish the peer review history of their article (what does this mean?). If published, this will include your full peer review and any attached files.

Reviewer #1: No

Reviewer #2: No

Reviewer #3: Yes: Dr Kamran Zaman

Figure Files:

Data Requirements:

Reproducibility:

References

---

## [Editor Report · Decision Letter 1]

2 Feb 2024

Dear Dr. Mukhopadhyay,

We are pleased to inform you that your manuscript 'Prevalence and changing antimicrobial resistance profiles of Shigella spp. isolated from diarrheal patients in Kolkata during 2011-2019' has been provisionally accepted for publication in PLOS Neglected Tropical Diseases.

Best regards,

Sivanantham Krishnamoorthi

Guest Editor

Stuart Blacksell

Section Editor

---

## [Editor Report · Acceptance letter]

14 Feb 2024

Dear Dr. Mukhopadhyay,

We are delighted to inform you that your manuscript, "Prevalence and changing antimicrobial resistance profiles of Shigella spp. isolated from diarrheal patients in Kolkata during 2011-2019," has been formally accepted for publication in PLOS Neglected Tropical Diseases.

Best regards,

Shaden Kamhawi

co-Editor-in-Chief

Paul Brindley

co-Editor-in-Chief
